# An Overview of the Strategies for Tin Selenide Advancement in Thermoelectric Application

**DOI:** 10.3390/mi12121463

**Published:** 2021-11-27

**Authors:** Rosnita Md Aspan, Noshin Fatima, Ramizi Mohamed, Ubaidah Syafiq, Mohd Adib Ibrahim

**Affiliations:** 1Solar Energy Research Institute (SERI), Universiti Kebangsaan Malaysia, Bangi 43600, Selangor, Malaysia; p102638@siswa.ukm.edu.my (R.M.A.); dr.noshinfatima@ukm.edu.my (N.F.); ubaidahsyafiq@ukm.edu.my (U.S.); 2Department of Electrical, Electronics and System Engineering, Faculty of Engineering and Built Environment, Universiti Kebangsaan Malaysia, Bangi 43600, Selangor, Malaysia; ramizi@ukm.edu.my

**Keywords:** optimization, performance enhancement, Seebeck coefficient, strategies, thermoelectric, tin selenide

## Abstract

Chalcogenide, tin selenide-based thermoelectric (TE) materials are Earth-abundant, non-toxic, and are proven to be highly stable intrinsically with ultralow thermal conductivity. This work presented an updated review regarding the extraordinary performance of tin selenide in TE applications, focusing on the crystal structures and their commonly used fabrication methods. Besides, various optimization strategies were recorded to improve the performance of tin selenide as a mid-temperature TE material. The analyses and reviews over the methodologies showed a noticeable improvement in the electrical conductivity and Seebeck coefficient, with a noticeable decrement in the thermal conductivity, thereby enhancing the tin selenide figure of merit value. The applications of SnSe in the TE fields such as microgenerators, and flexible and wearable devices are also discussed. In the future, research in low-dimensional TE materials focusing on nanostructures and nanocomposites can be conducted with the advancements in material science technology as well as microtechnology and nanotechnology.

## 1. Introduction

Many vital concerns related to environmental degradation and depletion of energy resources provide renewable energy as the best approach to solve these problems. The approach reduces the use of fossil fuels that are responsible for the crisis. Given the demand for clean energy with high performance, there is a need to develop renewable green energy instruments to address the energy problems and prevent further environmental pollution [1,2]. Solar panels, fuel cells, thermal generators, and wind turbines have significantly been considered in the last several decades and have shown reliable power generation efficiency. Besides the aforementioned renewable energy technologies, TE is among current renewable energy technology that is quite attractive due to the technology’s ability to directly convert between heat and electrical energy without moving components, fluids, or emitting greenhouse gases. As a result, it provides paths to silent, durable, clean, and green technology [3,4,5]. TE generators are environmentally stable and mechanically efficient electronic devices [6]. In addition, TE devices have high potential for energy conversion technologies in many fields, including aerospace, biomedical, air conditioning, power generators, and cooling devices [7]. However, thermoelectric generators (TEGs) are still immature, and many scientists are putting their efforts in enhancing efficiency through material modifications and/or device optimizations.

Many scientists have been working on the development of TE materials in recent years. For operating temperatures, TE materials are commonly divided into three groups [4,8], namely (a) low or near ambient TE materials (300 K–500 K), bismuth telluride (Bi_2_Te_3_) and its alloys as current primary materials [9,10,11,12,13,14,15], (b) mid-temperature TE materials (500 K–900 K), mainly based on lead-chalcogenides alloys and their derivatives [16,17,18], and (c) with high operating temperature TE materials (>900 K) such as SiGe alloys [19,20]. Numerous approaches in synthesizing TE materials, such as melt and growth, microwave synthesis, mechanical alloying, and hydrothermal have been effectively explored, with improvement in TE performance. Furthermore, it has been discovered that the nanostructure effectively reduces thermal conductivity, resulting in a higher TE figure of merit [21]. From the research done, it is obvious that TE technology still has a lot of room for development. Thus, one of the main motivations in developing this technology is to improve the performance of TE conversion for a wider range of applications [22,23,24].

This review describes the characteristics of crystal, band structure, and anharmonic bonding of mid-temperature TE materials, focusing on tin selenide (SnSe) to understand the origin of its superior TE properties. The study viewpoints are investigated for effective strategies that can lead to further enhancement in the efficiency of SnSe. The methods used in the fabrication of SnSe are also briefly described. There is also discussion on SnSe applications in TE fields like flexible and wearable electronics and microgenerators. SnSe is a very promising alternative to toxic and rare TE materials like lead (Pb) and tellurium (Te) because tin (Sn) and selenium (Se) are abundant, and their figure of merit values, presented as *zT*, is considered higher for single crystal (2.2–2.6 at 913 K) and polycrystalline (3.1 at 783 K), respectively [25].

## 2. Thermoelectric Characteristics and Performance Evaluation

The efficiency of a TE material can be gauged by their dimensionless figure of merit (*zT*), defined as:(1)$$zT=\frac{S^{2}\sigma }{\kappa }T,$$
where, *S*, *σ*, *T*, and *κ* represent the Seebeck coefficient, electrical conductivity, absolute temperature, and total thermal conductivity, which is the sum of thermal conductivity of the lattice (*κ*_lat_) and electrons (*κ*_e_), as presented by Equation (2) and further elaborated in Equation (3) [2]:(2)$$\kappa =\kappa_{e}+\kappa_{\mathrm{lat}}$$
and
(3)$$\kappa_{e}=L\sigma T,$$
where *L* presents the Lorenz factor [2]. For metals, the Lorenz factor is a constant at 2.4 × 10^−8^ J^2^/K^2^C^2^ but varies in semiconductors with the carrier concentration.

The dimensionless *zT* considers how efficiently the material converts its energy from thermal to electrical or vice versa. Within *zT* lies the power factor (PF), that is defined as *S^2^σ*. PF determines the power output for a given temperature difference applied. Both the Seebeck coefficient and electrical conductivity are related to the charge carrier transport; hence, the carrier concentration plays a crucial role in increasing the power factor [26]. Since the lattice contribution always dominated, thermal conductivity is less dependent on the charge carrier concentration. Thus, the carrier concentration which provides a given material with the maximum power factor is typically similar to that which gives the highest value of *zT* [27]. Basically, by raising the numerator or decreasing the denominator, the *zT* value can be increased without any restriction. However, the complex connections between TE parameters are prohibited from optimizing the *zT* value and TE conversion efficiency [8,17,28].

Consideration of the existence of both *κ*_lat_ and *κ*_e_ in total thermal conductivity, presents a new way of presenting *zT*, given by Equation (4):(4)zT=S2L1+κlatκeT

Equation (3) shows that with the decrement in the *κ*_e_ value, there is a decreases observed in *σ* too. Therefore, the idea to decouple *σ* and *κ*, to achieve maximum *zT* value, can be accomplished by minimizing the *κ*_lat_ value. After all, when the materials exceed the condition *κ*_lat_/*κ*_e_ << 1, the ideal situation arises, and the Seebeck coefficient alone defines the value of *zT*.

## 3. Background of Mid-Temperature Thermoelectric Materials

Mid-temperature Cu_2_Se- based [29,30,31,32,33,34,35], SnSe- based [36,37,38,39,40,41,42], PbTe- based [43,44,45,46,47,48,49,50,51] and GeTe-based materials [52,53,54,55,56,57,58], as summed up in Table 1, have been recorded with *zT* values close to 2 or higher. In general, better mid-temperature TE performance in Group-IV chalcogenides was observed. *zT* values have now surpassed 2 and have surged to 3 [25], with novel synthesis methods, optimization approaches, microstructures, and concepts have emanated, all of which encourage further advancement in the research field. However, they contain expensive germanium (Ge), rare Te, and toxic Pb, restricting their large commercial and domestic usability [3]. Alternatively, tin is a better choice than these high costs, atypical, and hazardous materials.

The tin telluride (SnTe) crystalline structure is similar to lead telluride (PbTe). Moreover, its band structure and transport properties are very alike too. In addition, the main disadvantage of SnTe is its chemical defect. As compared to PbTe in p-type or n-type, SnTe is often tin deficit. Tin-defect causes an increase in the carrier concentration in SnTe that limits the Seebeck coefficient and power factor. It is also difficult to achieve effective doping for TE applications. Recently SnSe created plenty of study opportunities in TE material fields due to its fascinating properties, such as very low thermal conductivity, substantial lattice anharmonicity, chemical stability, abundant availability on Earth, and being environmentally friendly [36,59,60].

### SnSe

SnSe is a semiconductor with an environmentally safe composition and one of the most promising and fascinating materials in the realm of TE. SnSe-based materials have recently been identified as outstanding TE materials due to their ultralow *κ*_lat_ due to the anharmonic scattering mechanism caused by lone s-pair electrons, which has been observed in a variety of chalcogenides [61,62] and the Cu ion liquid-like behavior in Cu_2_Se [63].

**Table 1 micromachines-12-01463-t001:** State-of-art thermoelectric properties of mid-temperature thermoelectric materials.

Year	Material Based	Composition	Carrier Type	Temperature (K)	Seebeck Coefficient (μVK^−1^)	Electrical Conductivity (Scm^−1^)	Thermal Conductivity (Wm^−1^K^−1^)	Lattice Thermal Conductivity (Wm^−1^K^−1^)	PF (*S2σ*) (μWcm^−1^K^−2^)	Max *zT*	Ref.
2017	Cu_2_Se	Cu_2_Se + 0.75wt.% C	p	1000	302	98	0.37	0.223	8.9	2.4	[20]
2017	Cu_2_Se	Cu_2_Se + 1% CuInSe_2_	p	850	150	550	0.4	N/A	12.4	2.6	[30]
2017	Cu_2_Se	Cu_1.98_S_1/3_Se_1/3_Te_1/3_	p	1000	243	182	0.57	0.297	10.7	1.9	[31]
2018	Cu_2_Se	Cu_2_Se + 0.15wt.% graphene	p	870	180	270	0.32	N/A	8.7	2.44	[32]
2019	Cu_2_Se	Cu_2_Se + 0.10wt.% carbon-coated boron	p	1000	300	100	0.4	0.250	9.0	2.2	[33]
2020	Cu_2_Se	Cu_2_Se + 0.8wt.% carbon nanodots	p	973	290	108.7	0.45	0.25	9.1	1.98	[34]
2020	Cu_2_Se	Cu_2_Se + 0.60wt.% C-Cu_2_Se	n	984	200	270	0.5	N/A	13	2.5	[35]
2021	Cu_2_Se	Cu_2_Se + 2wt.% carbon dots	p	880	100	302	0.5	N/A	3	2.1	[64]
2014	SnSe	SnSe single crystal (b-axis)	p	923	340	80	0.33	0.219	9.2	2.6	[36]
2016	SnSe	Hole doped SnSe single crystal (b-axis)	p	773	305	160	0.56	0.374	14.9	2	[37]
2018	SnSe	Sn_0.95_Se	p	873	320	70	0.3	0.208	7.2	2.1	[38]
2018	SnSe	Sn_0.98_Pb_0.01_Zn_0.01_Se	p	873	333	49	0.215	0.151	5.4	2.2	[39]
2018	SnSe	SnSe + 2% SnTe	p	793	280	195	0.58	0.348	15.3	2.1	[40]
2019	SnSe	Sn_0.99_Pb_0.01_Se + Se quantum dot	p	873	410	31	0.23	0.189	5.2	2	[41]
2020	SnSe	SnSe_0.95_ + 2% PbBr_2_	n	770	−480	25	0.3	0.25	8	2.1	[42]
2021	SnSe	Na_0.03_Sn_0.965_Se (purified)	p	783	280	115	0.2	0.07	9	3.1	[25]
2021	SnSe	SnSe + 3% CdSe nanoparticles	p	786	330	55	0.2	0.14	6	2.2	[65]
2013	PbTe	Pb_0.98_Na_0.02_Te + 6% MgTe	p	823	305	265	1.02	0.584	24.7	2	[43]
2014	PbTe	(PbTe)_0.86_(PbSe)_0.07_ + 2% Na	p	800	270	355	1.05	0.482	25.9	2	[44]
2014	PbTe	PbTe_0.7_S_0.3_ + 2.5% K	p	923	300	160	0.60	0.305	14.4	2.2	[45]
2015	PbTe	PbTe_0.8_S_0.2_ + 3% Na	p	923	240	300	0.82	0.266	17.3	2.3	[46]
2016	PbTe	Pb_0.98_Na_0.02_Te-8% SrTe, non-equilibrium	p	923	285	280	0.83	0.313	22.7	2.5	[47]
2017	PbTe	Na_0.025_Eu_0.03_Pb_0.955_Te	p	850	230	400	0.80	0.120	21.1	2.2	[48]
2018	PbTe	Pb_0.953_Na_0.040_Ge_0.007_Te	p	805	250	400	1.2	0.8	28	1.9	[49]
2019	PbTe	Bi-doped PbTe/Ag_2_Te	n	800	−250	300	0.6	0.3	15	2	[50]
2020	PbTe	Pb_0.96_Na_0.04_Te	p	860	260	416.7	1.3	0.7	29	1.9	[51]
2021	PbTe	Na0.03Eu0.03Pb0.94Te0.9Se0.1	p	850	255	377	0.8	0.4	25	2.3	[66]
2018	GeTe	Ge_0.86_Pb_0.1_Bi_0.04_Te	p	600	285	370.4	0.75	0.417	30.1	2.4	[52]
2018	GeTe	Ge_0.76_Sb_0.08_Pb_0.12_Te	p	800	260	469.5	1.1	0.537	31.7	2.3	[53]
2018	GeTe	Ge_0.89_Sb_0.1_In_0.01_Te	p	773	250	580	1.25	0.577	36.3	2.3	[54]
2018	GeTe	Ge_0.86_Pb_0.10_Sb_0.04_Te	p	600	260	476.2	0.92	0.491	32.2	2.1	[55]
2019	GeTe	Bi_0.05_Ge_0.99_Te	p	650	250	714.3	1.45	0.754	44.6	2	[56]
2020	GeTe	BiI_3_-doped Sb*_2_*Te_3_(GeTe)_17_	p	723	260	500	1.15	0.575	34	2.2	[57]
2020	GeTe	Ge_0.92_Cr_0.03_Bi_0.05_Te	p	600	225	666.7	1.1	0.5	32.5	2	[58]
2021	GeTe	Ge_0.92_Sb_0.04_Bi_0.04_Te_0.95_Se_0.05_	p	700	200	980	N/A	0.25	39.2	2	[67]
2021	GeTe	Ge_0.9_Mg_0.04_Bi_0.06_Te	p	700	255	1000	1.8	0.7	55	2.5	[68]

## 4. Characteristics of SnSe

The TE group ignored SnSe because of its high electrical resistivity (between 10^1^–10^5^ Ω cm [59], with a bandgap of 0.86 eV [59,69,70]. SnSe only received attention from researchers in 2014, when along with a particular crystallographic direction, Zhao et al., recorded the *zT* value of approximately 2.6 at 923 K, along the b-axis of a single crystal, which is a new record in bulk materials [38]. In 2018, a research by Chang et al. demonstrated that the Br-doped n-type SnSe crystals had a *zT* of 2.8 at 773 K along the a-axis [71]. According to recent research by Zhou et al., hole-doped SnSe polycrystalline samples made with properly purified reagents and stripped of tin oxides had approximately *zT* of about 3.1 at 783 K [25].

### 4.1. Crystal Structure

SnSe displays an assembly of layered orthorhombic arrangements with a, b, c lattice parameters as 11.49 Å, 4.44 Å, and 4.135 Å, respectively, forming the normally largest axis of a bi-planar layered structure [72,73,74]. There are three adjacent Sn atoms covalently bonded with each tin atom, and so on. SnSe possesses a Pnma space group at room temperature (No. 62). It underwent a phase transition at above 800 K with the van der Waals bonding of layers along the a-axis, where SnSe crystallizes in a higher symmetry (Cmcm space group No. 63), reducing its TE performance. SnSe transport properties demonstrate strong anisotropy [36]. The chemical bonding and particular anisotropic crystalline structure in SnSe enable in-plane electric transportation. The strong anharmonic characteristics severely impede thermal transportation in any direction, particularly outside the plane [75].

Figure 1 shows the crystalline structures alongside the a, b, and c axial directions of SnSe. The SnSe mono crystal cleaves effortlessly along the (001) plane due to its layered structure and phase transition from Pnma to Cmcm space group. This contributes to poor mechanical features [76]. Therefore, polycrystalline SnSe is preferred for applications as compared to single-crystal SnSe. A related study also concentrated on the polycrystalline SnSe, whereby most researchers presented materials as p-type conductors [39], while some presented n-type polycrystalline SnSe [77].

### 4.2. Band Structure

SnSe studies on the optoelectronic properties and bandgap (*Eg*) show its applications in the optical field, i.e., as photovoltaic (PV) and photodetector [79]. Most IV–VI compounds have a narrow bandgap. Literatures disclosed that bulk SnSe has a direct *Eg* of 0.9 eV and indirect *Eg* of 1.30 eV, respectively. As a consequence of quantum confining, the direct bandgap of bulk is less than that of thin films [80,81]. Besides the *Eg* parameter, electronic structures, density of state (DOS) structure of the bands, and effective mass of SnSe-based materials, are strongly linked to their TE performance [82,83].

The material band structures define the available electron energy levels, which are usually available due to the periodicity of lattice within the reciprocal space. These electrical transport properties of Fermi levels adjacent band structures can be determined directly. The work on TE materials from the band structure perspective showed that both the microscopic and macroscopic extremes of electrical transport are with the corresponding model case [84].

The calculated SnSe band structure was recorded in many density functional studies [36,85,86,87,88]. Shafique et al. [89] and Sirikumara et al. [85] calculated the band structure of bulk SnSe through first-principle density functional theory (DFT), and the values were 0.86 eV [89] and 0.62 eV at the Γ point [85], respectively.

Pletikosic et al. [90], while calculating the band structure of SnSe, observed valence band with highly anisotropic behavior, formed a variety of valleys, which are then rapidly dispersed in layers and are not spread across. For TE transport, the study reported features of the pronounced anisotropy low-temperature SnSe band structure which responded distinctively along the three axes of the crystal. The valence band has multi-valley hole pockets and supports less-effective, highly mobile charge carriers.

### 4.3. Anharmonic Bonding

The lattice vibration property is known as anharmonicity, governing its interaction and heat conduction phenomenon [73]. Sn-Se anharmonic bonding is one of the essential features of tin selenide crystal structure as it possesses ultralow thermal conductivity [91].

Figure 2 illustrates the anharmonicity and harmonicity. Φ(r) represents the potential energy, a_0_ is the lattice parameter, and the space between two neighboring atoms is presented by r [59]. The harmonicity shows a balance of phonon transport, and the anharmonicity shows the non-periodic transport of phonons. For harmonicity, force applied on the atom directly relates to its distance from the equilibrium position [82,92], and its proportionality constant is known as stiffness or spring constant. On the contrary, for anharmonicity, the spring constant depends on the displacement of the atom, being significant whenever a pair of phonons run between them [92,93]. The existence of the primary phonon varies the constant values of the secondary spring phonon, thereby becoming a channel with changed properties of elasticity [59]. Higher anharmonic behavior provides an increase in the phonon pairs scattering, which, as a result, decreases the lattice thermal conductivity without affecting the electronic properties of solid’s [59,82,92].

Meanwhile, all bonding is anharmonic in real materials. However, its degree varies for different materials. Grüneisen parameter (γ) is the anharmonic strength measuring factor that can be calculated by Equation (5) [92]:(5)$$\gamma =\frac{3\beta BV_{m}}{C_{v}},\;$$
where *β*, *B*, *V_m_*, and *C_v_* represent the coefficient of volumetric thermal expansion, modulus of the isothermal bulk, molar volume, and the isochoric specific heat per mole, respectively. An ideal harmonic crystal has zero Grüneisen parameter.

However, any positivity or negativity in the element value suggests its anharmonicity [36,59,93]. Zhao and team [36] recorded that the average Grüneisen parameter values of SnSe along the a-axis was 4.1, the b-axis was 2.1, and the c-axis was 2.3. The value of the Grüneisen parameter varies with the three mentioned axes, suggesting that SnSe is highly anharmonic [3,82,94]. The weaker bond between SnSe slabs transport of phonon. Hence, the high value of the Grüneisen parameter might be due to the soft bonding of SnSe, which ends in an extremely low lattice thermal conductivity [36]; hence, proving it to be the right candidate for good TE material.

## 5. Synthesis of SnSe

As mentioned earlier, the SnSe monocrystalline structure has fascinating TE properties. However, due to its poor mechanical properties, expensiveness, and rigid crystal growth conditions, the single crystals are restricted in their industrial-scale production [63]. Many researchers have concentrated on fabricating polycrystalline SnSe to avoid the slow process of single crystal growth. Melting and mechanical alloying are the two widely practiced ways to produce the polycrystalline SnSe, particularly in the case of melting routes.

In the following section, some commonly used methods for producing single crystal and polycrystal of SnSe TE materials were reported, i.e., Bridgman method, temperature gradient method, mechanical alloying method, hydrothermal/solvothermal (HT/ST) method, spark plasma sintering technique (SPS), and hot-pressing (HP) technology.

### 5.1. Bridgman Method

Bridgman method is considered an ancient technology for the growth of single crystals from a melt. However, it can also be used for polycrystalline ingot solidification. The techniques involve heating above the melting point of the polycrystalline material and gradually cooling it from the seed crystal end of the container. The same crystallographic orientation single crystal is cultivated on the seed and gradually forms along the length of the container. The method may be horizontally or vertically done, and it typically includes a revolving crucible/ampoule to stir the melt [95]. Figure 3 shows a Bridgman vertical furnace. The vertical Bridgman method usually produces a crystal of more excellent quality than the horizontal [28].

Firstly, the raw materials are weighed and placed into a quartz ampoule with sufficient stoichiometry. Then, the ampoule is evacuated, or an inert atmosphere is applied to the ampoule and later sealed. In a furnace, the materials are melted at a reasonable temperature and heating speed. The targeted temperature can be for the melting route above the melting temperature of SnSe, which is 1134 K. In the case of solid-state reaction, the temperature should be kept below its melting point. The speed of thermal annealing should be adjusted for appropriate Sn and Se reactions to form SnSe alloy. After the ampoule is maintained at a sufficient temperature to create a compound between the elements (from 12 h to several days), the oven is slowly cooled to room temperature. Finally, the vessel is taken out from the furnace, obtaining a single crystal ingot. The ingots obtained are crushed into powder and prepared for single crystal growth.

In research by Jin et al. [96], SnSe monocrystals of 100 mm length, 50 mm width, and 15 mm height were developed using the Horizontal Bridgman method. Polycrystalline SnSe was placed in a sealed crucible under a vacuum within the quartz crucible. The quartz crucible was put in a furnace which consisted of five separate heating areas at a horizontal angle of 30° to fabricate large crystals. Then, all zones were heated to 910 °C for 2 h. The SnSe crystal growth temperature gradient was 2 °C/cm–3 °C/cm and the crystallization time was almost 240 h. All areas were eventually cooled at room temperature for 60 h.

In a modified Bridgman technique, which is the Stockbarger method, the furnace is taken into two zones, often separated by a shelf. This separates bi-coupled furnaces with temperatures above and below the freezing point and creates a steeper temperature gradient between zones. Notice that for a vertical oven, the temperature gradient can be changed either by adjusting the heating element density or conventionally. The Bridgman-Stockbarger technique can optimize the sum of loaded elements into the ampoule, temperature gradient and decrease the temperature speed of the ampoule [28].

### 5.2. Temperature Gradient Growth Method

The temperature gradient method is used to produce 2D single-crystal semiconductors by using the same melting principle. This technique uses convective phenomena and the differences in heating element density to form a temperature gradient. A schematic diagram of the vertical furnace temperature gradient is shown in Figure 4. In this technique, a temperature controller controls the heat. Sn and Se powders will first be mixed and placed into quartz ampoules for the growth process, whereby the ampoules are then evacuated and sealed. The ampoule is often enclosed in another evacuated large quartz ampoule to avoid air oxidation of the sample if the inner ampoule breaks. The ampoules are placed into a vertical furnace and gradually heated at a high temperature over the melting point of SnSe. The ampoule is held at a high temperature for a long time to complete the reaction. Then, ampoules are continuously cooled to the melting point of the compound and quickly decreased to room temperature.

After the melting process is completed, annealing is often executed, followed by the process of cooling down to room temperature to improve the hardness and ductility of the final SnSe obtained. Furthermore, the annealing leads to a realignment of crystalline domains and an increase in the Seebeck coefficient and electrical conductivity [97].

In research by Zhao et al. [98], stoichiometric amounts of high purity Sn, Se, Ag, Ge were mixed and sealed under vacuum (<10^−4^ Torr) in quartz tubes. The tubes were gradually heated to 1223 K within 12 h and held for 12 h, then cooled down to 793 K and held for 6 h. The tubes were left in the furnace to cool to room temperature. The ingots were pulverized and then sintered with a spark plasma sintering method. Meanwhile, in a study by Yang et al. [99], the Sn, Se, and NaCl weighted compositions were mixed and placed into vacuum-evacuated, sealed quartz tubes. Then, an induction furnace was used to heat the quartz tubes for 10 min at approximately 800 °C. It was held for 10 min and gradually cooled to room temperature. The ingots obtained were hand-ground into a fine powder and sintered by the rapid hot-pressing method (RHP).

### 5.3. Mechanical Alloying

Mechanical alloying, particularly in ball milling (BM), has received extensive interest in synthesizing nanoparticle powders [100] since it is simple. A homogenous SnSe product is produced by blending the SnSe powder precursors with a high-energy ball mill [82]. In BM, elemental powders or intermetallic compounds are placed into a container of balls made of stainless steel, tungsten carbides, or zirconia. During the process, powders are milled in the argon atmosphere to prevent oxidation. The powders would be subjected to various ball-to-ball and ball-to-wall collisions [101]. The powders are continuously cold-welded and shattered during ball milling, resulting in nanostructured domains [102]. Moreover, mechanical alloying has often been considered a process after melting.

For example, BM was performed to synthesize n-type polycrystalline SnSe with titanium (Ti), Pb co-doping by Li et al. [76]. The milling was carried out for 6 h at 450 rpm with a powder-to-balls mass ratio of 1:20 in a planetary ball mill. In a 10 mm-diameter graphite mold, the BM-derivative powders were subsequently sintered at a uniaxial pressure of 50 MPa by SPS at 773 K for 5 min.

### 5.4. Hydrothermal/Solvothermal Method

Another route for SnSe successful methods to obtain SnSe products, including SnSe crystal nanorods [103,104] and nanosheets [105], is the hydrothermal (HT) method. The HT reaction occurs under high-pressure and high-temperature conditions, usually above the boiling temperature of the water, to attain a high vapor pressure and satisfy particular critical requirements. Synthesized products are then subjected to a post-treatment process involving isolation, washing, and drying. In the presence of an aqueous solution, this process blends precursor materials in a particular end-product stoichiometry with an aqueous solution, which is then loaded into an autoclave and sealed. The mixture is then pressurized and heated to a temperature above the solvent critical point for chemical reactions. It is retained for some time and then brought down to room temperature [85]. Water is used as a solvent during a chemical reaction. One of the most important properties of water is its dielectric constant. The dielectric constant of water decreases with an increase in temperature and pressure [82,106]. After autoclavisation, the stock is washed with distilled water to eliminate the remaining soluble salts and oven-dried.

For example, Pb-doped SnSe samples are successfully synthesized by using the HT method by Tang et al. [107]. SnCl_2_·2H_2_O, PbCl_2,_ and Se powders were used as raw materials. PbCl_2_ powder and SnCl_2_·2H_2_O were placed in a beaker and dissolved in deionized water. The solution was magnetically agitated for 10 min. NaOH was dropped into the mixture and then stirred for 10 min. The solution was moved to an autoclave. After Se powder was added, the autoclave was heated at 403 K for 36 h. Deionized water and absolute ethanol were used for many washes of synthesized powder after being cooled to room temperature. Finally, the black powder was vacuum-dried for 4 h at 333 K. The products achieved by the HT method have perfect crystal growth, controlled crystal size, high quality, and more cost-effective raw materials [63].

Solvothermal (ST) method is similar to the HT method. The difference is that it is synthesized at relatively high temperatures, and the solvent is not aqueous [108]. The method is also similar to HT for ST SnSe crystal synthesis. Similar to HT, post-treatments may be required after synthesis. The only difference is that centrifugation should be considered for the synthesized SnSe products to wash away any salt and ion in the solution by deionized water and ethanol to extract unreacted solvents and organic by-products [63]. Shi et al. [109] used a simple ST method to synthesize Sb-doped SnSe microplates by using Na_2_SeO_3_, SnCl_2_·2H_2_O, ethylene glycol (EG) anhydrous, Sb_2_O_3_, and NaOH as precursors. SnCl_2_·2H_2_O and Na_2_SeO_3_ were dissolved in EG Then, Sb_2_O_3_ and 10 mol L^−1^ of NaOH were added and stirred for 10 min at room temperature. The solution was then sealed in an autoclave. The autoclave was heated in an oven at 230 °C for 36 h. Then, it was gradually cooled to room temperature. The synthesized products were collected by centrifugation and repeatedly washed with ethanol and deionized water before drying in the oven at 60 °C for 15 h.

The benefit of HT and ST is that the synthesized products do not require a particular high synthetic temperature (generally below 500 K) [110] and are of high quality. Another benefit of the HT/ST method is that the morphology of the end product can be monitored by merely inserting a templating ligand [101,111,112]. The ligand is a chelating agent with reduced ionic precursors, which results in uniquely coordinated geometries that produce different morphologies [101,113].

### 5.5. Post-Treatment Synthesis

#### Sintering

The synthesized powders should be consolidated into the densified small pellets for further characterization. Three methods typically used to densify the nanostructured materials are cold pressing (followed by sintering), HP, and SPS [101]. Since the polycrystalline SnSe showed a low TE performance using the cold pressing technique, HP and SPS are two of the most commonly utilized methods for fabricating polycrystalline SnSe materials.

HP is a common method used for producing dense polycrystalline materials. Throughout the process, the powder is loaded into a mold, heated at a relatively high temperature enough by Joule to stimulate sintering, creep, and a flow of materials by pressure [28,114]. The theory of HP is to apply heat and mechanical pressure simultaneously. During this process, resistance heating and induction heating may generate Joule heating. The benefits of induction heating are independence between pressing and heating and the option of liquid or powder phases, even at low pressure. The drawbacks are uneven thermal distribution when the mold is positioned off-center and the lower heating rate, which is also influenced by the thermal conductivity of mold and the material induction characteristics. A heating element is used in indirect-resistance heating technology to heat the mold by radiation and convection phenomena. Figure 5 shows the schematic diagram of the polycrystalline SnSe sintering process by HP. In the case of SnSe, the HP technique allows the SnSe powder to be compactly sintered at high pressure and high temperature, which is up to 600 MPa [115]. For example, Li et al. [116] used the HP method to sinter the SnSe doped with Zn at 673 K in a 15 mm-diameter tungsten carbide die in a vacuum under a pressure of 600 MPa for 1 h.

SPS technique is a novel and modern sintering process known as the best technique for processing TE polycrystalline materials. Generally, SPS is regarded as a transformation to HP. The furnace is replaced by the mold, which contains the sample, heated directly through and ultimately through the sample by a current flowing through it. SPS is different from the conventional HP, whereby the heat generation is internal during the SPS method, and external heating elements will supply heat during the HP method.

Figure 6 shows a schematic of the SPS process. The technique enables bulk materials from powder to be processed during a short sintering time (0 min–10 min) at such a high heating rate (1000 °C/min). The powder is placed into a mold with two punches. During the SPS process, microscopic electric discharges are generated in the inter-particle gaps by applying a high electric pulsed current on the electrodes, induced plasma, and at the beginning of sintering. Once a spark occurs in the gaps between particles, the temperature may be high, and the melting will happen locally [117]. The molten fluid is then sputtered to the surfaces of the neighboring particles at high speed, thus connecting them. The discharge condition is relaxed when the connections are formed. Then, the local temperature reduces rapidly, and micro discharge stops, contributing to a partly melted micro-structure [118].

### 5.6. Advantages and Disadvantages of Synthesis and Post-treatment Synthesis Methods

From the literature discussed, many methods are used to fabricate SnSe crystals, including synthesis and sintering. Most of these methods require moderate and high temperatures to produce high-quality SnSe crystals for TE applications. However, each method has its pros and cons. Table 2 summarizes the advantages and disadvantages of each method that was described in the literature. The details can be obtained from the cited references.

## 6. Strategies to Improve the Thermoelectric Performance of SnSe

The performance of a TE materials is dependent on the electrical conductivity, thermal conductivity, and Seebeck coefficient of the material. Enhanced *zT* values are generally the result of high power factor values combined with low thermal conductivity. Moreover, a high power factor comes from a high electrical conductivity and a high Seebeck coefficient. Furthermore, electrical conductivity is dependent on carrier mobility and carrier density, which is determined by the electronic structure. Meanwhile, the Seebeck coefficient depends on the gradient of DOS of the conduction band near the Fermi level [86]. The lattice thermal conductivity could be significantly suppressed through proper modulating of the material microstructure [133]. However, the electronic thermal conductivity still depends very much on the electronic structure [86]. Consequently, several methods were investigated and divided into three: increasing power factor, reducing thermal conductivity and retaining a high power factor, and reducing thermal conductivity to improve TE performance.

For TE materials, texturing and doping are two significant methods of enhancing the TE performance of polycrystalline SnSe [82]. Even though single crystals are thermally conductive than polycrystals [134], SnSe, in its single crystalline form, is not ideal for TE devices because of high production costs, poor cleaving properties, and special requirements of the crystal-growth process [135]. Thus, polycrystalline SnSe is a popular alternative candidate for practical applications because it is machinable and scalable [109]. The highest *zT* in pristine polycrystalline SnSe is considerably below its single crystalline counterpart. Polycrystalline samples have low carrier mobility, high anisotropic nature, and high thermal conductivity, due to the presence of Sn oxides [136,137]. Many researchers have concentrated on increasing the performance via form transitions, metal doping, and hole doping [138]. Doping is shown to be an efficient way to optimize the power factor by tuning the carrier concentration. Furthermore, doping can cause point defects and nano-precipitates to reduce lattice thermal conductivity [139].

### 6.1. Enhancing the Power Factor

The undoped SnSe exhibits a carrier concentration of approximately 10^17^ cm^−3^ [36], significantly lower than the optimum TE material (between 10^19^ and 10^21^ cm^−3^) [20,140]. Both intrinsic and extrinsic factors affect the TE materials’ power factors. The intrinsic factors include doping elements solubility and band structure profiles near the Fermi stage. However, it is more apparent how power factors can also be improved via the extrinsic route by improving sample quality and optimizing the carrier concentration [2]. Then, chemical doping is used to increase carrier concentrations and boost SnSe TE performance.

The relation between the electrical transport parameters, electrical conductivity, and the Seebeck coefficient is complicated [141]. Developing high-performance TE materials while maintaining low thermal conductivity depends on enhancing or maintaining of power factor. A high power factor means that the system can generate large voltage and current. However, successful methods to increase the Seebeck coefficient ultimately reduce the electrical conductivity and vice versa. For example, electrical conductivity determines the potential of the material to produce electricity. It shall be written as Equation (6):(6)σ=nqμ
where n is carrier concentration, q is electric charge, and μ is carrier mobility. Successful σ needs high concentration and high mobility carriers. Hence, the σ can be enhanced by increasing these two parameters independently or simultaneously [142]. However, the Pisarenko relation indicates that the Seebeck coefficient is inversely proportional to n^2/3^. DOS effective mass (m*) or this large carrier increases the Seebeck coefficient but impedes carrier mobility, resulting in low electrical conductivity. Generally, two or more methods should be implemented concurrently to optimize electrical transport property [141].

Furthermore, strategies to improve the phonon scattering by microstructures control will usually deteriorate to a certain point of electronic efficiency (S^2^σ) due to the enhancement of electrical carrier scattering [2,143]. The tuning of carrier concentrations is the most common approach to optimize power factors since either low or high carriers cannot sustain high power factors because of the reversed S-to-σ relation [2].

Table 3 lists a few research performed with strategies to improve power factors. Some research has preferred sodium (Na) to be doped with SnSe. Na doping could effectively boost the carrier concentration, resulting in an improved electrical conductivity and increased power factor [127,144,145,146]. In research by Wei et al. [146], as compared to lithium (Li) and potassium (K), Na has the highest doping efficiency, although they are all elements in the same alkali metal group. Na was found to increase the Hall carrier concentration with a significantly higher Seebeck coefficient at 267.2 μV/K and contributing to a power factor at 5.8 μWm^−1^K^−2^ at 800 K. The electrical conductivity was increased after doping with 1 at.% of Na, which increased in *zT* value of 0.8. It showed that various doping elements would act differently [142]. Similarly, in this research, SnSe doped with K, Na, or Li, still showed a higher power factor as compared to undoped SnSe.

Li et al. [116] doped SnSe with zinc (Zn) and explored the effect of Zn doping on TE properties of SnSe. The results showed that a high *zT* of 0.96 at 873 K was obtained for Zn_0.01_Sn_0.99_Se, 41% higher than pure SnSe (*zT* = 0.68). The value was derived from an increased Seebeck coefficient and a high electrical conductivity.

Silver (Ag) has also been chosen by some researchers to be doped with SnSe to increase the carrier concentration of TE material and, in turn, increase the power factor. Leng et al. [147] recorded that at room temperature, the carrier concentration increased enormously with the addition of Ag content from 3.9 × 10^17^ cm^−3^ to 1.9 × 10^19^ cm^−3^ for Ag_0.01_Sn_0.99_Se. Chien et al. [148] found that Ag doping would effectively increase the carrier concentration from 0.5 × 10^18^ cm^−3^ to 9 × 10^18^ cm^−3^. The 3% Ag doping exhibited a maximum *zT* of 0.8 at 850 K, about 40% higher than the pristine SnSe. The consequence is mainly attributed to the enhancement of carrier concentration and power factor by Ag doping.

Chang et al. [149] showed that polycrystalline SnSe n-type could hit a high *zT* value of approximately 1.2 at 773 K. High performance is driven by the improved electrical conductivity caused by bromine (Br) doping and Pb alloy and low thermal conductivity of SnSe. The group suggested that Br is an effective doping agent for optimizing carrier concentration in n-type polycrystalline SnSe. A maximum *zT* about 0.54 was recorded in PbBr_2_-doped polycrystalline SnSe samples at 793 K, which benefited from an increased Seebeck coefficient and less affected electrical conductivity. These results contributed to a high power factor of 4.8 mWcm^−1^K^−2^ [126].

Kucek et al. [128] observed that thallium (Tl) is an effective SnSe dopant which brought *zT* to 0.6 at 725 K. They discovered a huge increase in hole concentration, which was why the electric conductivity was significantly increased. The *zT* parameter was boosted as a consequence of increased electrical conductivity. Sassi et al. [150] recorded a maximum *zT* of 0.5 at 820 K for p-type polycrystalline SnSe, suggesting that high *zT* values could result from proper optimization of SnSe transport properties.

### 6.2. Reducing the Thermal Conductivity

TE performance can also be enhanced by reducing thermal conductivity. The intricate relation between TE parameters accentuates the lattice thermal conductivity as the only parameter independent of carrier concentration. Consequently, reducing lattice thermal conductivity is an effective strategy to gain high TE performance [92,141].

Table 4 summarizes the effect of thermal conductivity reductions and enhancement of *zT* for SnSe by dopants. Optimally, to minimize the lattice thermal conductivity that can be accomplished by combining many forms of scattering mechanism, a full-spectrum phonon scattering strategy for heat-carrying phonons across several frequencies is required [2]. One of the most effective lattice thermal conductivity reduction approaches is all-scale hierarchical architecture [141]. It is successfully implemented in many TE material systems with high lattice thermal conductivity, such as tin chalcogenides and lead chalcogenides. In the sample Cu_0.01_Sn_0.99_Se, the lattice thermal conductivity was 0.2 Wm^−1^K^−1^, and *zT* approximately 1.2 at 873 K [152]. The hydrothermal synthesis was used to prepare the polycrystalline SnSe samples. The behavior of mesoscale grains and nanoscale precipitates interpreted all-scale hierarchical architectures to scatter phonons. Furthermore, atomic point-defect scattering of copper (Cu) played a crucial part in minimizing the lattice thermal conductivity.

Besides that, efforts were made to minimize thermal conductivity by introducing phonon scattering by dislocation, point defects, boundaries/interfaces, and resonant rattlers. Point defect scattering is one of the most effective phonon scattering techniques for reducing the lattice thermal conductivity [2,153]. This is based on the fundamental argument of physics that phonons are more prone to be scattered by point defects than electrical carriers, reducing the effect on carrier mobility [2]. The mechanism for reducing thermal conductivity through point defects is due to large mass and significant strain variations between host and guest atoms. Such fluctuations can be enhanced by creating vacancies, using interstitial atoms, or substituting atoms in TE materials [153]. As a result of the mass and strain fluctuation of polycrystalline n-type SnSe with Ti and Pb co-doping, the lattice thermal conductivity was significantly reduced [76].

Besides, exploring crystals with low lattice thermal conductivity is another technique in achieving high *zT* materials like lattice with good anharmonicity, low-symmetry, complex structure, and liquid-like lattice [154]. Many TE materials, such as SnSe, are highly anharmonic and have extremely low thermal conductivity [2,155,156]. With layered structures characteristic of the van der Waals force, two-dimensional materials have also been used because of their high TE performance and low lattice thermal conductivity. Taking advantage of layered structures, SnSe is showing promising TE performance [2]. SnSe doped with 2 at.% of Cu had a *zT* of 0.7 at 773 K. Average *zT* could be due to the SnSe intrinsic nanostructure formation integrated with the presence of the second phase of Cu_2_Se and ultralow thermal conductivity [4].

Comparing single crystals and polycrystalline SnSe, polycrystalline SnSe exhibits higher apparent thermal conductivity and significantly lower *zT* values. The presence of surface tin oxides in polycrystalline samples is responsible for their high thermal conductivity. Lee et al. [159] used an oxide-removal strategy that involves a chemical reduction process at 613 K under a 4% H_2_/Ar atmosphere to remove oxides. They discovered that the combination of ball milling and chemical reduction process efficiently removes the tin oxide layers from the surface of powdered SnSe-based materials. As a result, the thermal conductivity of polycrystalline hole-doped SnSe alloyed with 5% lead selenide is reduced to 0.11 Wm^−1^K^−1^, which is even lower than that of single crystals, and the *zT* increases to 2.5 at 773 K. According to recent research by Zhou et al. [25], hole-doped SnSe polycrystalline samples made with properly purified reagents and stripped of tin oxides, had an average *zT* of about 3.1 at 783 K. Its lattice thermal conductivity is about 0.07 W m^–1^ K^–1^, far lower than that of single crystals. As a result of these findings, a new age of high-performance practical TEs may begin to emerge.

### 6.3. Important Correlation of Power Factor and Lattice Thermal Conductivity

The individual and synergetic methods are of great importance to boost the electrical and thermal properties of TE materials [160]. Table 5 summarizes the synthesis methods and approaches used to simultaneously increase power factors and reduce thermal conductivity in SnSe materials. New concepts like band-alignment and nanostructure with compositional alloys offer an intelligent way of reducing lattice thermal conductivity while retaining a high power factor. In 2019, the power factor of polycrystalline SnSe was strengthened by Ge alloying and Ag doping to 1 mWm^−1^K^−2^. Consequently, a peak *zT* value of approximately 1.5 was reached at 793 K. The carrier concentration was increased by Ag doping, while the Seebeck coefficient was increased by Ge alloying due to enhancing the effective band mass and reducing the bandgap. Furthermore, Ge alloying greatly decreased lattice thermal conductivity via scattering phonons due to point defects [98].

In samarium (Sm) doped SnSe polycrystalline for the composition Sn_0.97_Sm_0.03,_ the integration of power factor enhancement and thermal conductivity reduction was seen. Dopant Sm can improve electrical conduction by raising its carrier concentration at room temperature to 12.71 × 10^16^ cm^−3^ from 3.65 × 10^16^ cm^−3^ for Sn_0.97_Sm_0.03_Se. This leads to a power factor increment to 214 μWm^−1^K^−2^, with a value of *zT*_max_ of 0.55 at 823 K [161]. The doping approach also has been applied to the polycrystalline SnSe p-type doped with NaCl. The NaCl dopant increases the SnSe electrical conductivity and carrier concentration, with a maximum power factor of ~4.14 μWcm^−1^K^−2^ at 810 K is achieved.

Chlorine (Cl) helps in achieving a low lattice thermal conductivity of ~0.27 Wm^−1^K^−1^ at 810 K. A maximum *zT* value of 0.84 was achieved at 810 K for the sample of Na_0.005_Sn_0.995_SeCl_0.005_. Another group reported that the power factor was improved by adding the layered MoS_2_/graphene (MoS_2_/G) to the p-type SnSe. The maximum power factor was 4.68 μWcm^−1^K^−2^ with SnSe + 3.2 wt% MoS_2_/G. A low lattice thermal conductivity was obtained as a result of the layered MoS_2_/G due to phonons scattering at grain boundaries [129]. The maximum zT was 0.98 at 810 K.

The synthesis method has recently allowed the control of decreased graphene oxide (rGO) concentrations in the nanocomposites. With the implementation of just 0.3% rGO, lattice thermal conductivity was substantially reduced at a minimum of 0.36 Wm^−1^K^−1^ at 773 K. This was because of the numerous SnSe/rGO interfaces and SnSe grain boundaries that could disperse phonons effectively. An elevated carrier concentration supported the electrical conductivity at low temperatures, which resulted in a *zT* of 0.91 for SnSe/rGO-0.3 at 823 K [162]. When Ge-doped SnSe was synthesized by a hydrothermal method and SPS, 3 mol% Ge doping enhanced the carrier concentration, resulting in high electrical conductivity of 50 Scm^−1^ and a power factor of 5.1 μWcm^−1^K^−2^. In conjunction with the ultralow lattice thermal conductivity of approximately 0.18 Wm^−1^K^−1^, Chandra and Biswas [163] reported the *zT* value of 2.1 at 873 K.

Meanwhile, Hong et al. [164] recorded a high *zT* of 1.1 at 800 K in the p-type SnSe nanoplates alloying with TE through band engineering. Sintered pellets have improved TE performance significantly due to increased power factor in temperature ranges of 300 K to 800 K and reduced thermal conductivity at the same time. The enhanced power factor was due to the high texture and enhanced carrier concentration. The increased phonon scattering due to point defect caused the decreased thermal conductivity by replacing Se-sites in Sn. The composition was prepared by melting synthesis and high-pressure sintering techniques. Cu-doping p-type SnSe polycrystalline *zT* was stated to be 0.79 at 823 K [165]. The results showed that electrical conductivity increased with increased Cu content. All samples decreased their total thermal conductivity as the temperature increased and reached their minimum values at 773 K.

A lower lattice thermal conductivity and an increased power factor were obtained by alloying p-type SnSe crystals with Te [40], resulting in a *zT* value of 2.1 at 793 K. The findings showed that Te alloying increased carrier mobility by equalizing the bond lengths and sharpening the valence bands, whereas the Seebeck coefficient remained high due to its several valence bands. This obtained a high power factor of ~14 μWcm^−1^K^−2^ at 793 K. The Te alloy enabled the mechanism of atomic displacement and thus decreased the lattice thermal conductivity [40]. Cha et al. [39] suggested that the phase-separation and nanostructured approaches can increase the PF and significantly reduce thermal conductivity; hence, *zT* 2.2 at an 873 K in polycrystalline SnSe with Pb and Zn co-doped was a very high *zT* record in polycrystalline SnSe for the time. In practice, the thermal conductivity of the SnSe is highly sensitive to oxygen content during preparation [7]. Both the power factor and lattice thermal conductivity of p-type polycrystalline K_0.01_Sn_0.99_Se could be improved simultaneously and prepared by mechanical alloying and SPS [137]. Potassium doping has increased carrier concentration and reduced Sn oxides. The disappearance of Sn oxides and the formation of nano-precipitates within grains significantly dropped the lattice thermal conductivity. The experiment yielded a *zT* value of 1.1 at the temperature of 773 K.

SnSe samples doped with rhenium (Re) and chlorine were prepared by solid-state reaction and SPS procedures. The Cl doping concurrently enhanced the carrier concentration. It changed the carrier type of polycrystalline SnSe, and Re-formed Re nano precipitates, which helped decrease the thermal conductivity by introducing the point defect. The bulk sample Sn_0.97_Re_0.03_Se_0.93_Cl_0.02_ obtained a zT value of 1.5 at 793 K [170].

In the field of TE, significant efforts were made by researchers, given the potential improvements in materials and devices. Some approaches, such as atomic structures and synthesis methods, were discussed from a selection of representative literature to improve the figure of merit of TE further. Technological advancements allow new strategies to integrate the strong interaction of TE transport parameters and continuously increase the zT values from the previous minimum value to much greater than 2. Since there are no conceptual or limitations to the maximum zT value, considerably more outstanding performance can be anticipated for the next generation of TE materials and devices.

## 7. Applications

Single-crystal SnSe, allows for efficient heat-to-electricity conversion and vice-versa, suitable for applications such as microgenerators, heaters or coolers. However, it is just too bulky for applications that demand flexible, wearable devices. The manufacture of 2D flexible TEGs is made possible by the design and fabrication of SnSe nanocrystals. Typically, there are two main approaches to achieve this goal [63]: (1) by fabricating SnSe-based thin films using drop-casting processes and assembling them with flexible substrates, and (2) fabricating organic/inorganic composite films containing SnSe nanocrystals as fillers.

### 7.1. Thin Film

SnSe nanosheets are suitable raw materials for the fabrication of SnSe thin films. Rongione et al. [105] demonstrated a low-cost, scalable solution approach for producing a nanostructured SnSe thin film. Drop casting was used to deposit a thin film of SnSe nanosheets over a 1 cm^2^ area of copper tape substrate. Near the nanosheet interfaces, phonon scattering results in high thermal boundary resistance and an ultralow thermal conductivity of 0.09 Wm^−1^K^−1^. The produced SnSe thin film was coated on a polyethylene terephthalate (PET) substrate to achieve a certain level of flexibility. According to the result, the SnSe thin film may be easily applied on flexible plastic substrates and maintain its outstanding TE performance through 1000 bending cycles.

Heo et al. [175] designed highly textured and hole-doped SnSe thin films to further enhance the anisotropy and boost the TE performance of SnSe thin films. A high S^2^σ of ≈4.3 μWcm^−1^ K^−2^ was achieved at 550 K. The current SnSe ink solution could also be used to fabricate SnSe thick films and flexible TEGs using a more practical solution process.

### 7.2. Organic/Inorganic Composite

Inorganic fiber-based TEs outperform organic or inorganic/organic hybrid TEs in terms of TE performance and designability, which can be attributed to their appropriate electronic structures and low lattice thermal conductivity. Therefore, various low-dimensional TE materials, such as nano or micro-based fibers, have been used in flexible TE generators or miniatured TE devices.

A solution processing procedure was used by Ju et al. [176] to fabricate flexible TE composite films made of conductive polyaniline (PANI)-coated SnSe_0.8_S_0.2_ nanosheets (NSs) and polyvinylidene fluoride (PVDF). Incorporating a modest amount of carbon nanotubes (CNTs) into the composite film can improve its TE properties even more. The CSA-PANI-coated SnSe_0.8_S_0.2_ NS/PVDF/CNT composite film has a maximum TE power factor of 297 mWm^−1^K^−2^ at 400 K, which is much greater than the CSA-PANI-coated SnSe_0.8_S_0.2_ NS/PVDF composite film without CNTs.

Zhang et al. [177] used the thermal drawing procedure to produce an ultralong single-crystal SnSe wire with a diameter ranging from micro to nanoscale. This process begins by thermally drawing SnSe into a polycrystalline, very flexible, ultralong, and mechanically stable flexible fiber-like substrate. Then the SnSe core is recrystallized to single-crystal over the whole fiber using a CO_2_ laser. The CO_2_ laser taper procedure may produce flexible SnSe fiber with various diameters while maintaining its single-crystal properties. When placed on the human body, such materials woven with functional SnSe fibers allow covered areas to breathe. This facile and low-cost technique paves the way for fiber-shaped single-crystal materials in various applications, from 1D fiber devices to multidimensional wearable fabrics.

## 8. Conclusions and Outlook

The heat to electricity converting TE cells has tremendous potential for powering devices. However, the efficiency of a TE generator is suffering due to ineffectual TE material performance. The TE figure of merit is determined indirectly, emphasizing the Seebeck coefficient, electrical conductivity, and thermal conductivity. A high power factor and low thermal conductivity are the major factors affecting zT value, while a high power factor is caused by a high Seebeck coefficient and high electrical conductivity.

In short, this review looks at the structure and typical synthesis methods currently used to prepare single-crystal and polycrystal of SnSe. Besides, strategies for improving zT are briefly discussed, and several approaches, including increased power factor and a decrease in thermal conductivity, were investigated to improve TE performance. SnSe and associated TE material were drawn tremendous attention because of their extremely low thermal conductivity attributed to their excellent anharmonic bonding and high electrical transport performance as credible materials with maximum potential for the next generations TE materials. It is impressive that such an ultralow thermal conductivity can be achieved in a simple compound because it has low molecular weight, large unit cell, or complex crystal structure. Remarkably, such a high power factor can be obtained only with orthorhombic crystal symmetry in the medium bandgap semiconductor.

The discovery of SnSe was indeed a considerable achievement in the TE field and contributed to the exploration and improvement of high-performance TE materials with low thermal conductivity. There is a long way to search for further strategies to explore TE properties synergistically and define the underlying mechanisms. In the future, most studies in the area of low-dimensional TE materials using nanostructures and nanocomposites can be carried out by material science, micro, and nanotechnology developments.

## Figures and Tables

**Figure 1 micromachines-12-01463-f001:**
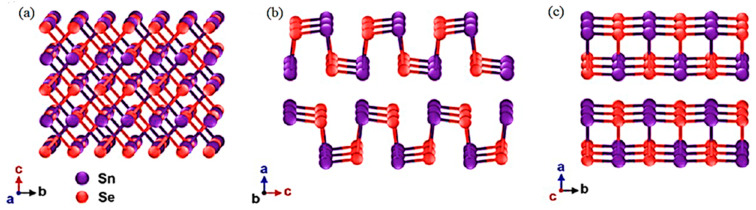
Crystal structures along with the (**a**–**c**) axial directions of SnSe. Reprinted with permission from [78].

**Figure 2 micromachines-12-01463-f002:**
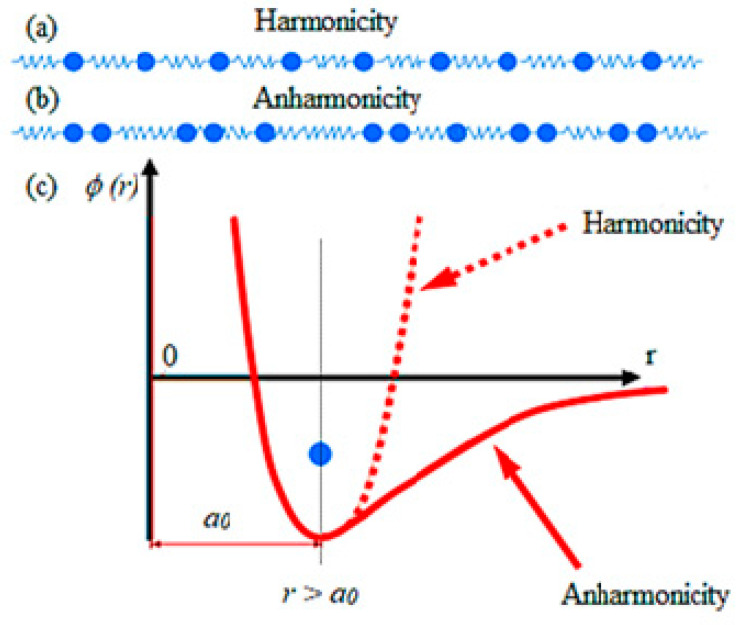
The schematic representations of (**a**) harmonicity and (**b**) anharmonicity. Meanwhile, (**c**) the harmonicity represents a balanced phonon transport, whereas the anharmonicity represents an imbalanced phonon transport. Reprinted with permission from [92].

**Figure 3 micromachines-12-01463-f003:**
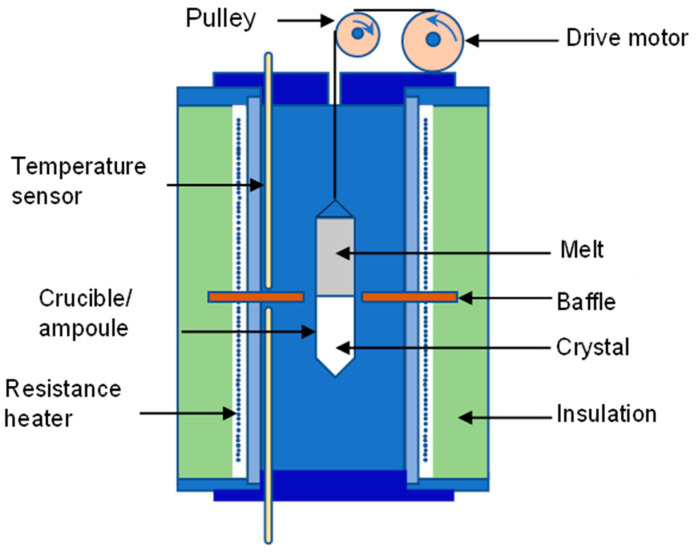
Bridgman technique for growing crystals.

**Figure 4 micromachines-12-01463-f004:**
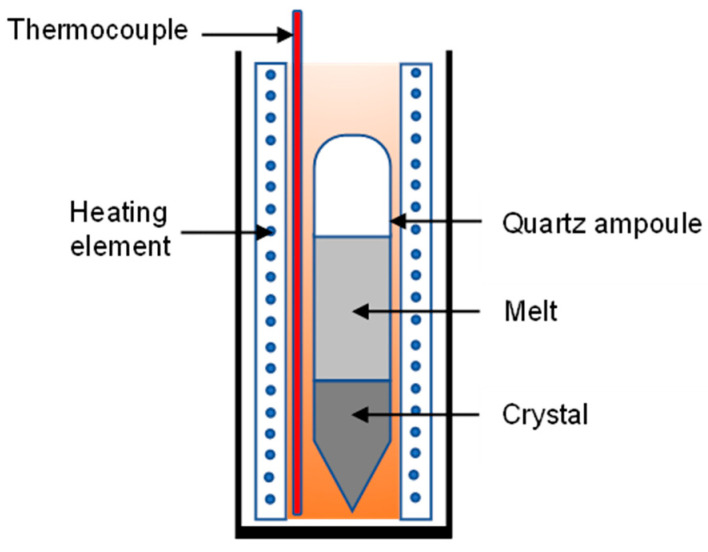
A schematic of a temperature gradient vertical furnace used in the temperature gradient growth method.

**Figure 5 micromachines-12-01463-f005:**
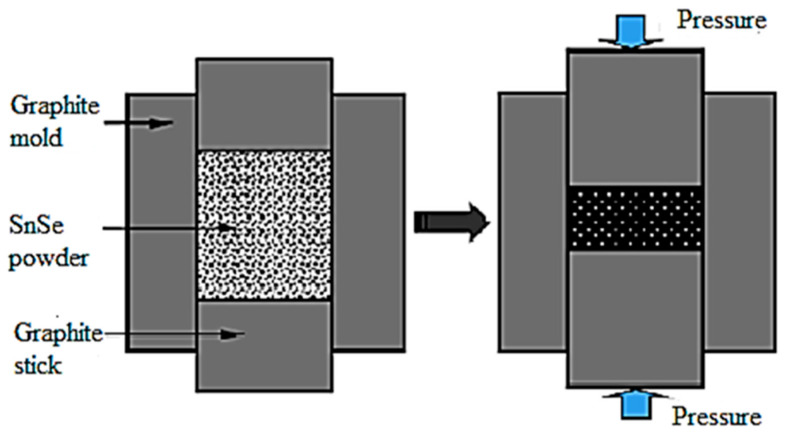
Schematic diagram of SnSe polycrystalline sintering by hot-pressing technique. Reprinted with permission from [96].

**Figure 6 micromachines-12-01463-f006:**
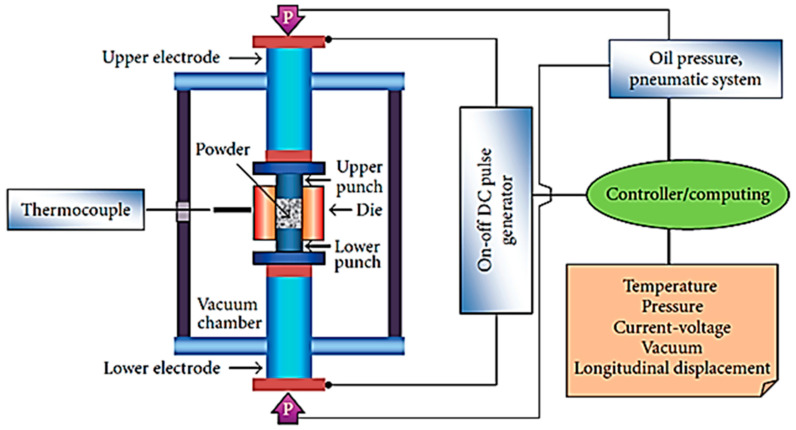
Schematic of the SPS process. Reprinted with permission from [115].

**Table 2 micromachines-12-01463-t002:** Summary of advantages and disadvantages of synthesis and post-treatment synthesis methods.

Method	Advantage	Disadvantage	Ref.
**Bridgman**	Technically, this method is simple [119]. Low cost [119].	During cooling, the compression of the solid by the container may result in stresses which are strong enough to dislocate the material [119].	[96]
**Temperature gradient growth**	Flexible [120].	Inhomogeneity of the products [120].	[98,99]
**Mechanical alloying (Ball milling)**	Useful for high purity nanoparticles with superior physical properties to be manufactured on a large scale [121,122]. It creates new and improved component properties that relay their grain size and material composition [121,122]. It could be used to produce alloys and compounds which are hard to manufacture through traditional melting and casting techniques and in-situ techniques [123].	It needs high energy [121]. Long milling time [121]. Powder contamination due to steel balls [121].	[76]
**Hydrothermal/solvothermal**	The synthesized products do not require very high temperatures of synthetic (generally below 500 K) [110]. The morphology of the end product can be monitored by simply inserting the templating ligand [101,113]. High yield, lower pollution, low energy consumption [124]. Environmentally safe and cheap [63].	The lack of stoichiometry control [101]. Materials can be hard to sinter [101,120].	[107,108,109]
**Hot pressing**	Pressure increases density at a specific temperature and can be achieved in shorter periods and at lower temperatures than conventional sintering [125]. Increase homogeneity in density, structure, and composition with increased processing stability [120].	The heating rate of HP is lower than SPS and has a longer holding time [120]. The equipment and tooling are more complex [125]. More costly than the sequential method of compaction followed by conventional sintering [125].	[116,126,127,128,129]
**Spark plasma sintering**	Lower sintering temperature and short holding times [120]. The cost of SPS is around 50–80% lower than conventional sintering methods [130]. The fast densification by SPS reduces the contact time between the sintering material and the graphite die and reduces the potential for unwanted product formation [130]. High energy efficiency [130]. Easy operation [131].	Such a short sintering time can fail to ensure the stability of the sample [120]. It needs an expensive pulsed direct-current (DC) generator [130].	[131,132]

**Table 3 micromachines-12-01463-t003:** TE properties and synthesis of SnSe with enhancement of power factor approaches.

Year	Composition	Synthesis Process	Approach	*T* (K)	*S* (μVK^−1^)	*σ* (Scm^−1^)	*κ* (Wm^−1^K^−1^)	PF (*S^2^σ*) (μWcm^−1^K^−2^)	*zT*	Ref.
2020	Polycrystalline SnSe	Mechanical alloying + SPS	Pressure applied during spark plasma sintering (SPS)	823				SnSe sintered at 120MPa has a PF of ∼3.9 μWcm^−1^ K^−2^	~0.7 with a pressure 60 MPa	[151]
2017	Na_0.02_Sn_0.98_Se	High-Pressure synthesis + SPS	Sodium doped polycrystalline SnSe with high-pressure synthesis (increased the hole concentration)	798	288.8	56.4	0.4	4.7	0.87	[144]
2017	Ag_0.03_Sn_0.97_Se	Surfactant-free solution growth process + SPS	P-type Ag-doped SnSe nanocrystals enhancement of carrier concentration and power factor by Ag doping	850	266.2	90.3	0.68	6.4	0.8	[148]
2017	SnSe_0.95_–3% PbBr_2_	Melting + hot pressing	N-Type polycrystalline SnSe by PbBr_2_ doping	793	−360	35	0.72	4.8	0.54	[126]
2017	Zn_0.01_Sn_0.99_Se	Melting + hot pressing	P-Type SnSe doped with Zn increased power factor coming from a high electrical conductivity and an enhanced Seebeck coefficient	873	328.5	74.1	0.73	8.0	0.96	[116]
2016	Na_0.015_Sn_0.985_Se	Melting + hot pressing	Na-doped p-type polycrystalline SnSe to optimize the carrier concentration	773	298.8	37.9	0.33	3.4	0.8	[127]
2016	Na_0.01_Sn_0.99_Se	Melting + SPS	Na_2_Se as an acceptor was doped into SnSe. Optimize the electrical transport properties, significantly to increase the carrier concentration	823	311.1	49.6	0.53	4.8	0.75	[145]
2016	SnSe_0.97_Br_0.03_ alloyed with Pb >10%	N/A	N-Type SnSe via Br doping and Pb alloying Br is an effective dopant to optimize the carrier concentration in n-type polycrystalline SnSe	773	390	30	*κ* = 0.37 *κ*_lat_ = 0.28	5.8	1.2	[149]
2016	Na_0.01_Sn_0.99_Se	Melting + SPS	Polycrystalline SnSe doped with three alkali metals (Li, Na, and K) Na has the best doping efficiency, leading to an increase in hole concentration	800	267.2	81.2	0.50	5.8	0.8	[148]
2016	Ag_0.01_Sn_0.99_Se	Melting + SPS	Polycrystalline Ag-doped SnSe compounds The carrier concentration was immensely enhanced	823	330.9	54.8	0.66	6.0	0.74	[147]
2016	Sn_0.995_Tl_0.005_Se	Melting + hot pressing	P-type polycrystalline Tl-doped SnSe	725	300	68.9	0.75	6.2	0.6	[128]

**Table 4 micromachines-12-01463-t004:** TE properties and synthesis of SnSe with reduction of thermal conductivity approaches.

Year	Composition	Synthesis Process	*T* (K)	*S* (μVK^−1^)	*σ* (Scm^−1^)	*κ* (Wm^−1^K^−1^)	PF (*S^2^**σ*) (μWcm^−1^K^−2^)	*zT*	Ref.
2021	Na_0.03_Sn_0.965_Se	Ball milling + chemical reduction	783	280	115	*κ*= 0.2 *κ**_lat_* = ~0.07	9	3.1	[25]
2020	Polycrystalline SnSe	Melting + SPS	823	250	25	*κ*= ~0.22 *κ**_lat_* = ~0.19	3.88	1.3	[157]
2020	0.5 wt.% graphene incorporated SnSe	SPS	823	270	27	0.18	2.2	1.06	[158]
2019	Na_0.01_(Sn_0.95_Pb_0.05_)_0.99_Se	Melting + ball milling + SPS	773	280	95	*κ* = 0.3 *κ**_lat_* = ~0.11	7.5	2.5	[159]
2018	Cu_0.01_Sn_0.99_Se	Hydrothermal + SPS	873	310	30	*κ* = 0.28 *κ**_lat_* = 0.2	3.5	1.2	[152]
2017	Sn_0.74_Pb_0.20_Ti_0.06_Se	Mechanical alloying + SPS	773	−450	16	0.55	3.0	0.4	[76]
2016	Sn_0.98_Cu_0.02_Se	Conventional fushion method + SPS	773	225	22	0.27	NA	0.7	[4]

**Table 5 micromachines-12-01463-t005:** TE properties and synthesis of SnSe by simultaneously increasing power factor and reducing lattice thermal conductivity approaches.

Year	Composition	Synthesis Process	*T* (K)	*S* (μVK^−1^)	*σ* (Scm^−1^)	*κ* (Wm^−1^K^−1^)	PF (*S^2^**σ*) (μWcm^−1^K^−2^)	*zT*	Ref.
2020	Sn_0.985_S_0.25_Se_0.75_	Mechanical alloying + SPS	823	350	40	0.38	~4.5	~1.1	[166]
2020	Sn_0.978_Ag_0.007_S_0.25_Se_0.75_	Mechanical alloying + SPS	823	325	50	*κ* = 0.24 *κ*_lat_ = 0.19	~5.3	~1.75	[166]
2020	SnSe/reduced graphene oxide(rGO)-0.3	In situ solution method + SPS	823	242	76	*κ* = 0.45 *κ*_lat_ = 0.4	5.3	0.91	[162]
2020	Sn_0.98_Na_0.016_Ag_0.004_Se	Melting + SPS	785	260	100	*κ*_lat_= 0.44	~ 7.3	~1.2	[167]
2020	SnSe_0.85_Te_0.15_	Hydrothermal + SPS	773	339	40	0.79	4.59	0.79	[139]
2020	Sn_0.97_Pr_0.03_Se	Mechanical alloying + SPS	773	−425	20	0.39	4.55	~0.9	[168]
2019	Sn_0.99_Na_0.01_Se–STSe	Melting + SPS	773	300	65	0.5	7	1.33	[169]
2019	Sn_0.97_Re_0.03_Se_0.93_Cl_0.02_	Melting + SPS	798	−450	31	0.38	6.0	1.5	[170]
2019	Ge doping (3 mol %) SnSe	Hydrothermal + SPS	873	260	50	*κ*_lat_ = 0.18	5.1	2.1	[163]
2019	Sn_0.99_Pb_0.01_Se_0.93_S_0.07_	Hydrothermal + SPS	873	320	37	*κ* = 0.18 *κ*_lat_ = 0.13	3.8	1.85	[171]
2018	SnSb_0.02_Se_0.96_	Solvothermal + SPS	773	−247	39.4	0.17	2.4	1.1	[109]
2019	Sn_0.90_Pb_0.15_Se_0.95_Cl_0.05_	Melting + SPS	823	−325	54	0.45	5.6	1.2	[6]
2019	Sn_0.975_Ag_0.01_Ge_0.015_Se	Melting + SPS	793	360	75	0.55	10	1.5	[98]
2018	Sn_0.93_Pb_0.02_Se	Hydrothermal + SPS	773	320	42	*κ* = 0.26 *κ*_lat_ = 0.2	4.25	1.4	[107]
2018	SnSe_0.9_Br_0.1_	Melting, mechanical alloying + SPS	773	−400	30	0.26	4.2	1.3	[172]
2017	Na_0.01_(Sn_0.96_Pb_0.04_)_0.999_Se	Melting + SPS	773	269.7	89.4	*κ* = 0.45	6.5	1.2	[173]
2017	(0.5% Na + 0.5% K)-co-doped SnSe	Mechanical alloying + SPS	773	374.7	34.9	*κ* = 0.32 *κ*_lat_ = 0.29	4.92	1.2	[136]
2017	SnSe_0.9_Te_0.1_	Solvothermal + SPS	800	322.8	57.4	0.44	6.0	1.1	[164]
2017	SnSe + 3.2 wt% MoS_2_/G	Melting + hot pressing	810	250	70	0.39	4.6	0.98	[129]
2017	Na_0.005_Sn_0.995_SeCl_0.005_	Melting + hot pressing	810	228.6	79.2	*κ* = 0.39 *κ*_lat_ = 0.19	4.1	0.84	[99]
2017	Sn_0.97_Cu_0.03_Se	Melting + high-pressure sintering	823	325.1	35.0	0.39	3.7	0.79	[165]
2017	Sn_0.97_Sm_0.03_Se	Melting + high pressure sintering	823	250.0	33.6	0.32	2.1	0.55	[161]
2017	Undoped polycrystalline SnSe	Hydrothermal + SPS	850	280	48	*κ* = 0.25 *κ*_lat_ = 0.19	4.0	1.3	[174]
2016	K_0.01_Sn_0.99_Se	Mechanical alloying + SPS	773	421.4	18.6	0.24	3.3	1.1	[137]

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
