# Peer review of "An Overview of the Strategies for Tin Selenide Advancement in Thermoelectric Application"

_micromachines, 2021, doi:10.3390/mi12121463_

Round 1

Reviewer 1 Report

This review paper summarized the development of SnSe thermoelectric materials including state-of-the-art ZT values, different sample preparation methods with pros and cons discussed, strategies to improve ZT of SnSe, and the correlations between PF and kappa lattice. The structure is well organized and manuscript well written. I would suggest adding some discussions on the thermoelectric device of SnSe and its potential application. The challenges to fabricate SnSe devices and possible solutions should be discussed. With adding this part, the manuscript can be accepted for publication.

Reviewer 2 Report

The work presents an updated review of the performance of tin selenide (SnSe) as a mid-temperature thermoelectric (TE) material. The characteristics, synthesis methods and strategies to improve the TE performance are evaluated.

The review paper is interesting and well structured. However, some improvements are still needed. My suggestions are given as below.

1. The Abstract is disorganized making it difficult to follow. It does not give sufficient key information to make it useful. Therefore, it does not catch enough interest. 

2. Same with the first part of the introduction, (from line 24 to 46). More organized and specific information of the research thematic is encouraged. 

Relative papers regarding SnSe background and thermoelectric application as the title of the review indicates is suggested to be added. 

3. In general language all over the manuscript needs a thorough review. This issue must be properly addressed

(ie. line 53: “ their figure of merit values, presented as ZT, is considered high” )

4. In Table 2. Summary of advantages and disadvantages of synthesis and sintering methods. I suggest to add an extra column with the references from literature that use the methods (even if they are mentioned in the text).

5. In Table 1. The State-of-art thermoelectric properties of mid-temperature thermoelectric materials are mentioned. Why the state-of-art Tin Selenide materials presented in the table  (ie. Ref [52]  with ZT 3.1, to name one) are not included in the “6. Strategies to Improve the Thermoelectric Performance of SnSe” tables? 
